# Physical Activity, Gut Microbiota, and Genetic Background for Children and Adolescents with Autism Spectrum Disorder

**DOI:** 10.3390/children9121834

**Published:** 2022-11-27

**Authors:** Julio Plaza-Diaz, Ana Mei Radar, Aiman Tariq Baig, Marcos Federico Leyba, Maria Macarena Costabel, Juan Pablo Zavala-Crichton, Javier Sanchez-Martinez, Alex E. MacKenzie, Patricio Solis-Urra

**Affiliations:** 1Department of Biochemistry and Molecular Biology II, School of Pharmacy, University of Granada, 18071 Granada, Spain; 2Children’s Hospital of Eastern Ontario Research Institute, Ottawa, ON K1H 8L1, Canada; 3Instituto de Investigación Biosanitaria IBS.GRANADA, Complejo Hospitalario Universitario de Granada, 18014 Granada, Spain; 4Department of Cellular and Molecular Medicine, Faculty of Medicine, University of Ottawa, Ottawa, ON K1H 8M5, Canada; 5Children’s Hospital of Eastern Ontario, Division of Urology, Department of Surgery, University of Ottawa, Ottawa, ON K1H 8L6, Canada; 6Faculty of Education and Social Sciences, Universidad Andres Bello, Viña del Mar 2531015, Chile; 7Escuela de Kinesiología, Facultad de Salud, Universidad Santo Tomás, Viña del Mar 2520298, Chile; 8PROFITH “PROmoting FITness and Health through Physical Activity” Research Group, Sport and Health University Research Institute (iMUDS), Department of Physical Education and Sports, Faculty of Sport Sciences, University of Granada, 18071 Granada, Spain; 9Servicio de Medicina Nuclear, Hospital Universitario Virgen de las Nieves, 18014 Granada, Spain

**Keywords:** children, autism spectrum disorder, intestinal microbiota, physical activity, genetics

## Abstract

It is estimated that one in 100 children worldwide has been diagnosed with autism spectrum disorder (ASD). Children with ASD frequently suffer from gut dysbiosis and gastrointestinal issues, findings which possibly play a role in the pathogenesis and/or severity of their condition. Physical activity may have a positive effect on the composition of the intestinal microbiota of healthy adults. However, the effect of exercise both on the gastrointestinal problems and intestinal microbiota (and thus possibly on ASD) itself in affected children is unknown. In terms of understanding the physiopathology and manifestations of ASD, analysis of the gut–brain axis holds some promise. Here, we discuss the physiopathology of ASD in terms of genetics and microbiota composition, and how physical activity may be a promising non-pharmaceutical approach to improve ASD-related symptoms.

## 1. Introduction

In general, autism spectrum disorder (ASD) refers to a group of disorders that affects the manner in which people interact with others, communicate, learn, and behave [1]. As a developmental disorder, ASD can be diagnosed at any age, but most cases manifest during the first two years of life [2]. Although ASD prevalence has been reported in a wide variety of studies with general estimates in the range of one in 100 children worldwide [3,4], a number of well-controlled studies have reported significantly higher figures. In particular, over the past decade, the number of ASD diagnoses in the United States has increased rapidly (currently one in 59 births, compared to one in 150 in 2000) [5]. Less is known of ASD prevalence in low and middle income countries due to a lack of data [6]. The male-to-female ratio of ASD has long been thought to be 4:1 [7] although a recent systematic review has found it may be closer to 3:1 [8]. One reason for this ostensible male predominance is the existence of an apparent diagnostic gender bias; girls who meet ASD criteria are disproportionately less likely to be diagnosed with the condition [8]. Environmental and genetic factors also contribute to a child’s likelihood of developing ASD [6], including sex-linked genetic factors (e.g., the X chromosome gene protective effect) and hormonal factors (e.g., prenatal hormones), which are proposed to attenuate the risk in females and increase it in males [9,10,11,12,13,14,15,16,17]. In addition, DNA variants including de novo mutations have a significant influence on the severity of the disorder [18]; a number of rare variants could play an influential role in ASDs [19]. Clinically, this is an important finding, as it may lead to genetic testing as a supplemental test with behavioral analyses to assist in an early detection of ASD. A number of disorders under the broader ASD diagnostic rubric are caused by rare, penetrant genetic variants [19]. A recent study analyzing whole-genome sequencing data derived from 3474 families revealed that ultra-rare variants can also lead to large-effect risk variations. This study reports and replicates a transmission disequilibrium of private, likely gene-disrupting variants in probands. However, 95% of the burden of mutation was found to lie outside of known de novo mutation-enriched genes [18].

In addition to DNA variants, there are several heritable and nonheritable risk factors associated with ASD, including increased postnatal head growth, brain volume anomalies, neuroanatomical defects in the amygdala, immune system dysregulation, serotonergic systems anomalies, and distinct neuropeptides or neurotrophins [20]. Putative neuropathological processes are proposed to begin in utero [20], including antenatal dysregulation of cortical layer formation and neuronal differentiation [21,22].

It is becoming increasing recognized that the mode of newborn delivery, vaginal or Cesarean section, can have a profound impact on both early and late outcomes in the life of a child, chiefly because of the significant differences observed in the newborn microbiome [23]. Infections, allergies, and inflammatory disorders are more likely to occur following Cesarean delivery [24,25,26,27]. ASD has been extensively studied in this regard, with many studies showing Cesarean section, when compared with natural birth, confers an increased risk of developing ASD [28,29,30,31], including a Canadian study showing a 1.23 times greater risk [32]. A 2016 cohort study of 5 million children in Norway, Sweden, Denmark, Finland, and Australia showed this number to be 1.26 for emergency or planned Cesarean sections [33]. However, such an association was not detected when sibling controls were used in another study, suggesting the existence of confounding familial genetic and/or environmental factors [34]. There is clearly a need for further investigation of the relationship between Caesarean delivery and ASD.

Children with neurodevelopmental disorders, including ASD, are frequently affected by gastrointestinal problems and gut microbiota dysbiosis [35]. Evidence from both animal models [36,37] and limited human studies [38,39] suggest the gut–microbiome-brain axis plays a regulatory role in both immune function and central nervous system function. Several studies have shown that certain bacteria belonging to the phylum *Bacteroidetes*, including *Barnesiella*, *Bacteroides*, *Parabacteroides*, *Prevotella*, *Odoribacter*, *Proteobacteria* (e.g., *Proteus*, *Parasutterella*), and *Alistipes*, are increased in the intestinal microbiota of ASD patients when compared to healthy individuals. On the other hand, levels of bacteria belonging to the phylum *Actinobacteria*, such as *Bifidobacterium* species, are often lower in ASD patients [40,41].

Physical activity is considered to be beneficial therapeutically in reducing inflammatory pathways [42]. It has been suggested that exercise may have a progressive effect on the composition of the intestinal microbiota of healthy adults [42,43]. For example, 12 weeks of structured exercise programs are more likely to have beneficial effects on the gut microbiome and to induce potential anti-inflammatory effects through changes in the gut microbiome and immunometabolic pathways [44].

A recent meta-analysis showed a beneficial impact of physical activity in children and adolescents with ASD, with significant improvement in social interaction, communication, motor skills, and reduced severity of the disorder [45]. Physical activity, however, did not have a significant impact on stereotyped behavior (i.e., the repetitive, meaningless, and aimless behaviors common to ASD patients [46]). Clearly, physical activity interventions may benefit children with ASD, in particular those involving continuous physical activity [45].

In the following sections we shall present an overview of physical activity, gut microbiota, and genetic targets as they relate to children and adolescents with ASD.

## 2. Physical Activity with a Special Emphasis on ASD

The term “physical activity” refers to any movement that is produced by skeletal muscles and requires energy expenditure. This can include leisure activities, occupations, education, household chores, home activities, and transportation [47].

According to the WHO general recommendation for physical activity, children should spend an average of at least 60 min/day of moderate-to-vigorous intensity over a week, including vigorous-intensity aerobic activities, and at least 3 days of strengthening activities for muscle and bone. Furthermore, the guidelines recommend reducing recreational screen time and limiting sedentary activities. In this regard, the latest WHO physical activity guidelines underscore the strong association between physical activity and improved cognitive, mental and physical health outcomes in children and adolescents [48]. Although the guidelines do not include evidence concerning ASD explicitly, it does outline the beneficial relationship between health outcomes and activity in children with general intellectual disability [48]. Conversely, the guidelines cite prolonged sedentary behavior as a risk for adverse health outcomes in children and adolescents living with disabilities. In children and adolescents, recreational screen time has a stronger association with adverse health outcomes than total sedentary time [49].

Recently, 24 h movement guidelines for children and youth aged 5–17 years have been published [50,51] with specific recommendations for physical activity (at least 1 h), screen time (no more than 2 h of screen time), and sleep (9–11 h of sleep (or 8–10 h for children aged 14 or older [50,51]. In the case of the ASD population, there has been a scarcity of research of this type that examines behaviors utilizing an integrated framework.

Interestingly, compared to children without ASD, those with the diagnosis engage in significantly less physical activity [52,53], have less sleep [54] and have more recreational screen time [55,56].

Analysis of 1008 youth with ASD included in the US 2016 National Survey of Children’s Health, 6.5% of children and 4.4% of adolescents follow 24 h movement guidelines ((a) physical activity, (b) screen time and (c) sleep duration), while nearly 12% of both do not adhere to any guidelines. The remaining percentages of children and adolescents with ASD met one of three guidelines, around 46% and 40%, respectively, and met two of three guidelines, 35.5% and 43.6%, respectively. In comparison to children without ASD, children with ASD were less likely to meet all three guidelines, and adolescents with ASD were less likely to meet the guidelines for physical activity and screen time [57]. Further analysis of data from the same survey revealed that only 2.6% of 746 adolescents with ASD met the guidelines. Contrary to this, 19.9% did not meet any guidelines [58]. Further, a recent study with 1165 young people with ASD from seven different countries found that only two percent of the participants adhered to all three guidelines for 24 h movement [59]. Specifically, only 7.2% reach the physical activity guidelines, 46.4% of the total sample met the screen time guideline, and 55.9% met the sleep duration guideline [59].

The distribution of physical activity among young adults with ASD taken from the US National Children’s Health Survey is summarized in Figure 1. Clearly, given its association with numerous chronic diseases and health concerns in the general population and specifically those with ASD [60], physical inactivity in ASD patients is an issue of concern [60].

In aggregate, studies have demonstrated associations with physical health, including physical fitness (muscular and cardiorespiratory fitness), bone health, cognitive outcomes, metabolic health, and mental health to physical activity [61,62,63]. Conversely, the decreased levels of physical activity observed in children with ASD are associated with being overweight or obese [61,62,64], poor quality of life [63,65], and poorer general health [59], all elements that have been previously identified as serious concerns in ASD patients [61,62,63,66].

### Physical Exercise Interventions for Children and Adolescents with ASD

An exercise program is a structured, planned form of physical activity designed to either improve or maintain physical fitness [42,67]. A number of mental health disorders have been managed or even treated successfully through physical exercise. The possible mechanisms proposed include inducing molecular changes impacting the anti-inflammatory state as well as other mechanisms affecting the peripheral and central nervous system [68,69]. Physical exercise is a promising non-pharmacological approach to improve several health aspects or alleviate symptoms in youth with ASD [70,71], including symptomatology and associated comorbidities [69] such as motor skills or sociability skills [72], and stereotypical motor behaviors [73]. Meta-analysis and systematic reviews have shown the positive effect of physical exercise interventions on reduction of stereotypical behavior [74], as well as reducing deficits in social interaction and executive function (particularly in terms of cognitive flexibility and inhibition of behavior) in children and adolescents with ASD [69,75,76].

Physical exercise interventions comprised of 10 to 90 min sessions lasting for 8 to 48 weeks have all been assessed [77,78] (Table 1 summarizes the data from physical activity in children and adolescents with ASD). While beneficial effects are seen, the optimal dosage (intensity, duration, and frequency of exercise) has not been determined and may vary between individuals [69].

Exercise interventions that have been studied include aquatic exercises [79,80,81], walking [82,83], cycling [84], yoga and dance [85,86], martial arts [87,88,89], horseback riding [90,91], fundamental movement skills [92], basketball [93], and football-based training [94]. Programmed aquatic exercises have been shown to improve sleep onset latency, sleep duration, physical fitness, water orientation, water-based skills, and social behaviors in children with ASD [79,80,81]. Walking and running programs appear effective for weight management and academic engagement in youth with ASD [82,83], while cycling promotes both self-regulation and sustained physical activity [84]. Yoga has shown a significant positive impact on classroom behavior [85] and, when combined with dance and music, to reduce behavioral symptoms, internalizing, externalizing, atypicality, and depression in youth with ASD [86]. In other interventions, martial arts classes have been shown to improve stereotypy and communication deficits [87,88] while horseback riding improves social–emotional functioning such as adaptative and executive functioning and irritability and stereotypic behavior [90,91]. Finally, a school-based program including fundamental movement skills (i.e., catching, throwing, jumping) reduced anxiety in children with ASD [93,94] while basketball- and football-based interventions improved sleep quality and executive function (including efficiency and duration) [93,94].

While exercise appears to provide numerous benefits for children and adolescents with ASD, currently, no specific guidelines or recommendations exist regarding the characteristics of physical exercise programs for these individuals. The design and implementation of physical exercise programs for children with ASD can present several challenges (for initial strategies to create these guidelines, see [95]) taking into account motor skills, social interaction, sensory processing, and environmental factors. As a consequence, special attention to the ASD-targeted physical exercise program’s environment is important in order to communicate and instruct exercise appropriate to the child’s level of interest and ability [95]. This includes consideration of environmental factors such as lighting, sound, and the size of the area when planning the program [96,97]. Effective methods for teaching individuals with ASD include visual supports (e.g., visual schedules, photos, video modeling) [69,71,98]. In addition, some exercises can be equally effective whether performed individually or in a group [73], while other investigators have shown individuals can find participating in group-based exercise challenging [99], suggesting exercise programs should be implemented according to the child’s preferences [71,73].

In general, every child/adolescent is different in terms of interest and capacity in exercise with no “one-size-fits-all” methodology to treatment. The tailoring of physical exercise programs to reduce barriers (e.g., non-flexible class options, social communication difficulties, overstimulating environments) and maximize the facilitators (e.g., adaptive equipment, peer volunteers, specially-trained staff) constitutes the optimal approach.

## 3. Gut Microbiota in Children and Adolescents with ASD

Microbiota is defined as the microorganisms (bacteria, fungi, and viruses, among others) which naturally inhabit a particular biological niche; in the case of humans, the gut microbiota is comprised of approximately 500–1000 species which collectively impact human well-being [100]. The makeup of the human gut microbiota is influenced by a large number of endogenous and exogenous factors including birth delivery method, host immune response, diet, use of antibiotics and other drugs, genetic characteristics of the host, infections, diurnal rhythms, and exposure to environmental microbes [101,102,103].

Exercise and physical activity have also been shown to change the composition of the gut microbiota, something which is associated with improved energy homeostasis and regulation (for a comprehensive revision, please refer to [42,104]) [42,104,105,106].

There are physiological connections and thus information exchange between the microbiota, the gut, and the brain [107]. An estimated trillion microorganisms are associated with the gastrointestinal tract, the body’s largest surface [108]. A key component of the system is the intestinal barrier, constituted of three primary components: commensal microbiota, mucus, and epithelial cells, closely interconnected by tight junctions [109,110]. Diet plays an essential role in shaping the microbiota throughout the life cycle **[111,112].** A burgeoning literature, including cross-sectional studies, has compared the microbiota of individuals affected with neurological disorders and those of healthy age-matched control subjects.

Given that antibiotic usage affects the health and composition of the gut microbiome [113] and the fact that children with ASD are frequently prescribed antibiotics due to their susceptibility to gastrointestinal infections [114], the study of the gut microbiota in ASD represents a compelling research topic. Indeed, the gut microbiome has been identified as a critical component in the pathogenesis of ASD in systematic reviews [115] with evidence of dysbiosis which may contribute to the development and severity of symptomatology [116]. ASD intestinal dysbiosis is characterized by persistently reduced alpha diversity (defined as mean diversity of species at different locations within a local scale [117], while a beta diversity can be defined as the ratio between the mean diversity of species at regional and local levels), the presence of immature microbes, an altered composition of 20 operational taxonomic units, reduced detection rates of taxon, and 325 metabolic functions that are deregulated [118].

A dysregulated microbiota composition has been linked to specific symptoms related to ASD in several studies [41,119,120,121,122]. Comparing ASD with controls, several common characteristics were identified in gut microbiota composition [123,124,125]; an overview of the many studies is presented here. In children with ASD, there is a minor relative abundance of *Streptococcus* and *Bifidobacterium* genera [126]. Pyrosequencing of gut microbiota has shown those with ASD, when compared to control cases, have greater levels of *Bacteroidetes* and lower levels of *Firmicutes* [114]. In contrast in another study, a significant growth in *Firmicutes*/*Bacteroidetes* ratio was observed in patients with ASD as a result of a reduction in *Bacteroidetes* [127]. Specifically, *Alistipes*, *Bilophila*, *Dialister*, *Parabacteroides*, and *Veillonella* decreased in relative abundance among the cohort with ASD, whereas *Collinsella*, *Corynebacterium*, *Dorea*, and *Lactobacillus* were increased [127]. Both *Actinobacterium* and *Proteobacterium* had higher levels in the ASD group compared with control cases. The prevalence of *Desulfovibrio* species and *Bacteroides vulgatus* in the stools of children with ASD is significantly higher than that children without ASD [114].

Previous studies have shown *Sutterella wadsworthensis* to be associated with gastrointestinal infections; interestingly, ASD children with and without digestive disturbances, have a significantly higher prevalence of *Sutterella* species [128,129,130].

In addition, there are higher numbers of *Ruminococcus torques* in children with ASD who have functional gastrointestinal disorders [130].

There is a greater relative abundance of *Bifidobacteraceae*, *Lactobacillaceae*, and *Veillonellaceae* phyla in the gut microbiota of children with ASD, whereas members of the *Prevotellaceae* phylum dominate the gut microbiota of healthy children [131]. *Bacteroides*, *Coprococcus*, *Akkermansia*, and different *Ruminococcus* species were more abundant and diverse in children with ASD [132].

A recent systematic review found that the presence of *Clostridium*, *Sutterella*, *Desulfovibrio*, and *Lactobacillus* was significantly greater in children with ASD, but the findings were inconsistent across studies due to the poor evaluation of external factors such as antimicrobial use, gastrointestinal symptoms, and diet [133].

There exists a strong correlation between constipation and bacterial patterns in subjects with ASD and neurotypical individuals [127]. This is possibly due to individuals with ASD having a higher level of bacterial taxa belonging to *Clostridium cluster* XVIII and *Escherichia*/*Shigella*. *Haemophilus parainfluenzae*, and *Faecalibacterium prausnitzii* abundances in the feces of children with ASD were also seen to be lower following multiple testing corrections [134]. The differences are not restricted to bacteria with a trend to higher *Candida* levels observed in ASD subjects compared to neurotypical subjects [127].

In addition, some studies have referenced changes in oral microbiota in children with ASD [135]. Pathogens such as *Haemophilus* and *Streptococcus* were found in increase concentrations in salivary and dental samples of patients with ASD while commensal bacteria such as *Selenomonas*, *Prevotella*, *Actinomyces*, *Fusobacterium*, and *Porphyromonas* were reduced. The presence of *Prevotellaceae* was significantly reduced in dental plaques from patients with ASD. Furthermore, bacteria belonging to the *Prevotella* and *Rothia* genera were also correlated with clinical indices, such as severity of the disease and oral health status (e.g., dental caries) [136].

### 3.1. Metabolomic Studies in ASD

Children with ASD are believed to have less integrated gut–blood barriers, allowing increased release of bacterial metabolites, such as indoles, short chain fatty acids (SCFAs), and lipopolysaccharides, triggering a variety of physiological reactions including severity of behavior, sleep, and gastrointestinal symptoms [137].

In one recent study, ASD was distinguished from neurotypical subjects by the presence gram-negative *Veillonella* and members of *Enterobacteriaceae*, as well as 17 microbial metabolites [118]. In another report, children with ASD and sleep disorders reported lower *Agathobacter* and *Faecalibacterium* levels in their microbiome while 3-hydroxybutyric acid, melatonin, and serotonin levels were increased, changes which may contribute to sleep problems and other core symptoms [138].

A systems-wide network analysis approach has been used to show children with ASD had lower levels of butyrate and lactate producers [139]. The direct measure of the stool from ASD has shown lower levels of both butyrate and acetic acid, and higher levels of valeric acid. Among individuals with ASD, there is a reduction in the abundance of butyrate-producing taxa, including *Rubinococcaceae*, *Lachnospiraceae*, *Eubacterium*, and *Erysipelotrichaceae*, as well as increases in valeric acid related to bacteria, known as Acidobacteria. An increase in *Barnesiella*, *Coprobacter*, *Fusobacterium*, and valeric acid related to bacteria (*Actinomycetaceae*) in constipated subjects with ASD has also been observed [140]. Children with ASD have also been found to have greater proportions of *Proteobacteria* and *Actinobacteria* at the phylum level, as well as *Bacilli*, *Actinobacteria*, *Erysipelotrichi*, and *Gammaproteobacteria* [141].

In addition, children with ASD also have significant deficiencies in microbial detoxifying enzymes and pathways when compared with neurotypical children. Further, biomarkers for mitochondrial dysfunction have been shown to correlate strongly with these deficiencies. Measurement of these detoxifying enzymes enabled the accurate identification of individuals with ASD from controls on the basis of clinical ratings [142]. Using deep metagenomic sequencing in fecal samples, ASD-related bacterial markers have been identified and persistent underdevelopment of the gut microbiota observed in children with ASD [143].

There is an extreme shortage of clinical trials investigating neurological functions, metabolites related to the integrity of the blood–brain barrier, neuronal energy metabolism, and neuroprotection. Due to the significant importance of these functions to brain homeostasis as well as the wide range of diet-derived metabolites detected, particularly amino acid- and polyphenol-derived metabolites, it is strongly recommended that future interventions focus on these pathways and compounds in order to promote brain homeostasis. There have been few studies demonstrating that nutritional or gut microbiome modification strategies can significantly improve neurological and psychiatric disorders [144].

Table 2 summarizes the information about children and adolescents with ASD and gut microbiota.

### 3.2. Microbiota–Gut–Brain Axis in ASD

An emerging literature has implicated the gut–brain axis in diverse aspects of the CNS including neuronal development, cognitive regulation, brain ageing and, to some extent, overall functioning of the brain. In this regard, the gut microbiota can transmit signals to the brain, releasing metabolites that can have impact themselves or modulate CNS levels of psychoactive compounds [145,146]. Several neurotransmitters involved in ASD are believed to be regulated by the microbiome, including glutamate, serotonin, and dopamine [147,148].

The neurotransmitter communication hypothesis in ASD pathogenesis invokes both hyper- and hypo-glutamine models at different stages of development. Previous studies have established that antagonists of NMDARs or AMPARs show some clinical benefit with ASD [149]. In addition, the excitatory glutamate pathway has been implicated in gut–brain communication and possibly ASD pathology. The glutamate pathway also plays a crucial role in cell adhesion that connects pre- and post-synaptic neurons, mediates trans-synaptic signaling, and shapes neural networks by specifying synaptic functions [150]. In this regard, in vivo neuroimaging of ASD individuals has demonstrated correlation between ASD phenotypes and glutamate/glutamine levels in various areas of the brain [149].

The T356M DNA variant in the SLC6A3 gene encoding the dopamine transporter is associated with ASD in both mice and humans. Mice homozygous for T356M weigh less and have a lower body fat percentage, as well as manifesting altered dopamine signaling and metabolic dysfunction. In addition, the oral microbiota of these mice are altered with lower *Fusobacterium* abundance, as well as glucose dysregulation. There was a positive association between *Fusobacterium* abundance and improved glucose handling as well as a decrease in body fat [151].

Recent studies have demonstrated that host genetics influence both the composition of the overall microbiome as well as the composition of individual bacterial species [152]. In Knight et al., changes in *Enterobacteriaceae* abundance were associated with Crohn’s risk variants within the NOD2 gene [153]. In a genome-wide association study of host genetics and microbiome in healthy individuals, it has been found that a loss-of-function variant in the fucosyl transferase 2 gene and a variant conferring hypolactasia near the lactase gene are associated with stool abundance of *B. longum* [154].

### 3.3. Modulation of Intestinal Microbiota as Treatment for Youth with ASD

As documented above, ASD is associated with an altered gut microbiome profile suggesting that modulation of gut bacteria may reduce ASD symptoms [155]. As outlined further below, human studies suggest that probiotics, prebiotics, and combinations thereof are beneficial in reducing ASD symptoms although more work remains in establishing this conclusively [156]. In general, multi-species probiotic supplementation, such as microbiota transfer therapy, appears more effective than single-species supplementation.

There is also evidence that intestinal dysbiosis contributes to systemic inflammatory response and gastrointestinal symptoms, which can modify permeability across the blood–brain barrier and the synaptogenesis process in the brain as a result of increased intestinal permeability [157]. One recent study showed that female mice fed a high fat diet have offspring with a dysbiotic gut microbiome, which could be causally related to both asocial behavior as well as deficits in synaptic plasticity [158]. In a more recent mouse study, maternal exposure to elevated levels of glyphosate was shown to result in abnormal composition of gut microbiota in juvenile offspring as well as ASD-like behavioral changes. These changes were linked to increased soluble epoxide hydrolase activity after the maternal exposure to glyphosate [159]. In a third example, the maternal immune activation mouse model was used (pregnant dams were injected with the viral mimic poly(I:C)), a model known to display ASD features. Microbiota alterations were documented in offspring, as well as increased intestinal permeability, with a resulting altered serum metabolomic profile. One metabolite in particular induced abnormal behavior in naïve mice. Importantly, oral treatment of offspring that exhibited ASD-like behavior with commensal B. fragilis both modulated the composition of microbiota and improved communication, stereotypic behavior, anxiety, and sensorimotor impairments suggesting that gut bacteria have effects on the host metabolome which in turn impacts behavior [36]. Another report evaluated the safety of the VISBIOME formulation (mainly *Bifidobacterium* and *Lactobacillus* strains) in children with ASD, eliciting only modest effects on gastrointestinal findings; this observation might be confirmed by studying a larger number of participants [160].

Another study using probiotics combined with fructo-oligosaccharides saw an increase in beneficial bacteria (*B. longum* and *Bifidobacteriales*) and diminution of *Clostridium*, with significant decreases in both ASD and gastrointestinal symptoms. Significantly lower levels of SCFAs, propionic acid, acetic acid, and butyric acid levels were observed in children with ASD, as well as a hyperserotoninergic state (increased serotonin) and dopamine metabolism disorder (decrease in homovanillic acid levels). After the administration of probiotics plus fructo-oligosaccharides, the levels of SCFAs reached those found in the control group along with associated decreases in serotonin levels and increases in homovanillic acid levels [161].

In a more recent study, the administration of probiotics resulted in significant improvements in gastrointestinal symptoms, maladaptive behaviors, communication skills, and perceived parental stress in children with ASD. Although no significant differences were observed in microbiome alpha diversity among groups, beta diversity in children with ASD were significantly different between start and at 3 months compared to those without ASD. There were some taxa that were positively associated with the three-month samples, such as *Bifidobacterium longum*, *Streptococcus thermophilus*, *Limosilactobacillus fermentum*, and *Ligilactobacillus salivarius* [162].

A report looking at probiotics, prebiotics, and synbiotics observed beneficial impacts on the metabolic activity and gut microbiota of children with ASD. Probiotic treatment led to the increase in *Lactobacillus* relative abundance. In contrast, the prebiotic treatment led to the growth of *Bifidobacterium* relative abundance and the reduction in *Lachnoclostridium* relative abundance. In the prebiotic and synbiotic treatments, changes in microbial metabolism were related to an increase in SCFAs concentrations and a decrease in ammonium levels [163].

A six-week prebiotic intervention on 30 children with ASD significantly reduced bowel movements and abdominal pain in children on exclusion diets. The decrease in abundance of *Bifidobacterium* species and *Veillonellaceae* was accompanied by a growth in abundance of *Faecalibacterium prausnitzii* and *Bacteroides* species. This group had a higher correlation between fecal amino acids and bacterial populations than those on an unrestricted diet. Following prebiotic intervention, a significant increase was observed in metabolites of the *Lachnospiraceae* family [164].

AB-2004 is a spherical carbon adsorbent with a high surface area that is highly effective in removing uremic toxins, including those produced by gut bacteria. After 8 weeks of treatment with AB-2004, children with ASD showed improvements in anxiety, irritability, and gastrointestinal health [165].

An in silico study modeling Western and high-fiber diets predicted greater reduction in microbial toxins produced in the gut using the latter diet. The authors also propose this effect would be augmented by the addition of probiotic bacteria such as *Bifidobacterium longum*, *Lactobacillus acidophilus*, *Akkermansia muciniphila*, and *Prevotella ruminicola* to promote gut microbiota balance and reduce oxidative stress in the gut and brain [166].

A recent urinary metabolomic ASD study demonstrated increased phenylalanine and decreased tyrosine levels, as well as higher bacterial degradation products, phenylpyruvic acid, phenylacetic acid, and 4-ethylphenyl-sulfate. A nine-metabolite profile distinguished the urine metabolomes of children with ASD from their unaffected siblings. In addition, a subset of ASD children with increased gut permeability were identified using the lactulose:mannitol test. In these children, four additional gut-permeable metabolites (phenylacetyl glycine, fucose, nicotinurate, and 1-methyl-nicotinamide) were found in their urine, which were not seen in the urine of the remaining children with ASD [167]. In another urinary metabolomic study, the authors identified significant dysregulation to the purine, tyrosine, phenylalanine and tryptophan pathways, characterized by increased phenylalanine and decreased tyrosine levels. As a result of bacterial degradation, the concentrations of phenylacetic acid, phenylpyruvic acid, and 4-ethylphenyl-sulfate were also elevated [168].

A recent systematic review of probiotic and prebiotic supplementation for children with ASD assessed different prospective and open-label designs [169]. Prebiotics when used alone improved gastrointestinal symptoms; when used in combination with an exclusion diet (casein and gluten free), significantly decreased antisocial behavior was observed. In contrast, there was limited evidence for any effect with probiotics. The authors concluded that, to date, there is limited evidence supporting the use of probiotics/prebiotics to relieve both behavioral and gastrointestinal symptoms in children with ASD. Finally, another systematic review found that prebiotics and probiotics did not significantly reduce the severity of ASD, gastrointestinal problems, and comorbid psychopathology in children with ASD [170].

In contrast, an open-label study with two-year follow-up treating patients with fecal microbiota transfer therapy (microbiota transplantation) proved relatively effective and safe, with significant reductions in gastrointestinal disorders and ASD symptoms as well as changes in the gut microbiota, and an increase in microbial diversity all observed [171]. The documented significant differences in plasma metabolites were to some extent corrected, approximating those found in healthy children [172]. As a result of microbiota transfer therapy, overall bacterial diversity also increased, with greater abundance of *Bifidobacterium*, *Prevotella*, and *Desulfovibrio*. The changes persisted for at least 8 weeks after treatment cessation [173].

Another report suggests the impact of microbiota transplant therapy on gut microbiota may be related, at least in part, to the presence of *Eubacterium coprostanoligene* [174]. In another study, a significant reduction in both constipation and abnormal feces, as well as in white blood cells and globulin levels, was observed after microbiota transplant therapy with a positive correlation between size effect and number of treatments [175].

The use of microbiota transplant therapy may thus be an effective therapeutic strategy in the treatment of both digestive and behavioral symptoms in patients with ASD, and also has shown favorable results in the treatment of certain infections, such as *C. difficile* [176].

Despite this wealth of data, ultimately, additional prospective cohort studies, possibly involving defined microbiological treatment of ASD, may be necessary to fully delineate the relationship between the microbiome and ASD and to identify the best therapeutic approach [177,178].

## 4. Genetics of ASD

GWAS, monogenic ASD disease gene cloning, and mouse modeling have all been employed to investigate the genetic basis of ASD [179]. Results of whole-genome screening have established the significance of synaptic functioning in ASD [180].

A number of genes involved in postsynaptic excitatory neuronal activity have been implicated in ASD; for example, copy number variations (CNVs) in the NLGN-NRXN-SHANK pathway and other synaptic genes such as DLGAP2 and SynGAP have been identified [181,182,183,184]. A network of interactions has been established between in high-risk ASD genes, including HOMER, SHANK, synaptic cell-adhesion molecules neurexin, neuroligin, and FMR1 [185,186,187]. Neurexin and neuroligin, which are important components of synaptic cell adhesion, have been associated with cognition and its disorders [188].

Patients with ASD have been found to have mutations in Shank2 and Shank1, with the loss or the addition of a copy both being observed [189]. Previous research has suggested that mutations in the gene encoding Shank3, an intracellular scaffolding protein linked to neuroligins via PSD-95 and GKAP, may also be common in ASD [188]. *NLGN3*, *NRXN1*, and *NLGN4* variants have also been observed in familial ASD. ASD patients with deletions of X-chromosomal DNA including *NLGN4* have been reported [179]. Other genes and their proteins are clearly involved in ASD pathogenesis: a genotyping of 12,000 individuals with ASD and subsequent analysis found 102 ASD risk genes, 72 of which were expressed in early excitatory neurons [190,191].

Researchers have been exploring the genetics of *DLG4*, a gene which encodes a postsynaptic scaffolding protein PSD-95 (Postsynaptic Density of 95 kD), a key constituent of the postsynaptic density [192,193].

The postsynaptic density (PSD) is an electron-dense region at the membrane of a postsynaptic neuron, juxtaposed with presynaptic active zone, which serves as a scaffold bringing receptors in proximity to the presynaptic neurotransmitter release sites [194].

Through the coordination of protein–protein interactions and excitatory synapses, PSD-95 plays a significant role in synaptogenesis and neural (synaptic) plasticity [195]. During synaptic maturation, PSD-95 interacts with, stabilizes, and traffics NMDARs/AMPARs to the postsynaptic membrane, thus playing a key role in synaptic plasticity, glutamatergic transmission, and spine morphogenesis [196]. Given PSD-95’s role in D1 dopamine receptor localization helping modulate NMDA currents [197,198], PSD-95 is thought to directly regulate NMDA function [199].

The modulation of synaptic function by PSD-95 is dependent on age and subregion [200]. As outlined above, there exists synaptic clustering of PSD-MAGUKs and glutamatergic receptors; this is thought to regulate hippocampal synaptic plasticity, transmission, and hippocampus-dependent behavior [201]. PSD-95 also binds to neuroligins through the third PDZ domain and to glutamate receptors of the AMPA type through its first PDZ domain [188]. Through its GK domain, PSD-95 interacts with GKAP, and the C terminus of GKAP binds to the Shank family of scaffold proteins, which is enriched in the PSD. Therefore, PSD-95 works as a major functional bridge connecting the neurexin–neuroligin–SHANK pathway implicated in ASD [202]. Ankyrin repeats are found in Shank, as well as SH3 domains, PDZ domains, SAM domains, and a proline-rich region [203]. It may also serve as a scaffold protein in the PSD by cross-linking NMDA or neuroligin/PSD-95 with regulators of the actin cytoskeleton [204]. The formation of synapses and the specification of synapse diversity may also be affected by synaptic cell adhesion molecules [205].

Furthermore, overexpression of Shank3 [206] increased PSD-95 levels, which were linked to an increase in the size, number, and strength of excitatory synapses [206].

Recently, a significant literature linking PSD-95 to learning and cognitive deficits observed in schizophrenia and ASD has emerged. There is evidence that PSD-95 deficiency alters the function and composition of NMDA/AMPA receptors in certain brain regions, with a possible pathogenic role in neuropsychiatric disorders [196]. DNA variants in genes involved in glutamatergic synapses including PSD-95 have been identified in psychiatric patients.

The social brain consists of the amygdala, the prefrontal cortex, the *nucleus accumbens*, the anterior cingulate cortex, the anterior insula, the hippocampus, and the temporal sulcus. Interestingly, social isolation and other anxiety-inducing triggers increase PSD-95 levels in the hippocampus and amygdala while decreasing PSD-95 in the frontal cortex [207]. PSD-95 dysfunction may contribute to synaptic malformations associated with neurological disorders by affecting synaptic plasticity at the dendritic spines [196].

The deletion of the DLG4 gene, which encodes PSD-95, leads to behavioral abnormalities that are similar to those found in ASD patients [202]. PSD-95 knockouts are predicted to result in an increase in silent synapses in prefrontal neurons, resulting in long-range disconnection from other areas of the brain [208]. According to Coley and Gao, 2019, PSD-95 KO mice displayed impaired memory and learning novelty and hyposociability during adolescence [202,208,209]. During development, a PSD-95 deficiency disrupts social brain and synaptic connectivity development, resulting in hyper/hyposociability [208].

In other genes, the loss or deactivation of FMRP is thought to cause ASD, resulting in increased expression of PSD-95 protein and an increase in AMPA recruitment to the postsynaptic membrane [196]. FMRP is responsible for the translation of receptors, kinases, and scaffolding proteins at the synaptic level. The dephosphorylation of FMRP increased PSD-95 translation via mGluR1-dependent pathways, and in the absence of FMRP, PSD-95 ubiquitination is inhibited, resulting in enhanced synaptic development [196].

In all, the potentially beneficial effect conferred by drugs which act on glutamate receptors, as well as genetic evidence of ASD involvement of glutamate receptors and other post synaptic density genes, and other interventions including gut microbiota modulation could be investigated as a potential therapeutic avenue [210,211].

With respect to ASD genetics outside of the synaptic protein realm, in one recent small study single nucleotide variations (SNVs) were identified in 26 ASD children and 26 matched controls using whole-exon sequencing. Significantly more SNVs were found in genes related to innate immunity, retrograde axonal transport, and protein glycosylation in people with ASD. In addition, these SNVs were associated with the composition of the microbiome and a broad spectrum of microbial functions, particularly metabolism. The authors postulate a causal relationship between the observed SNVs and microbial and metabolite abundance involved in the neurotransmitters’ metabolic network [152].

## 5. Conclusions

There is at least one child in 100 worldwide estimated to have ASD, with recent estimates pointing to an even greater prevalence. There now exists a wealth of published data showing that children with ASD are often affected with, in addition to gastrointestinal issues, significant gut microbiota dysbiosis along with an associated metabolomic dysregulation. The study of the gut–brain axis may thus significantly help understanding ASD disease physiopathology. This work can be viewed through the lens of the extensive genetic delineation of DNA variants impacting synaptic physiology as well as associated neurotransmission pathways in ASD. In the future, careful delineation of the role of interventions such as exercise and dietary use of pre and probiotics on the potential correction of both dysbiosis and metabolomic perturbation would be welcome. If such an effect is observed, it is hoped it may also bring some correction of anomalous ASD-associated neuronal signaling and thus clinical benefit for the millions affected by this common disorder. An increasing amount of research is being conducted on how host genetics interact with the gut microbiome [212]. This research is in the context of complex human diseases, with previous evidence coming primarily from animal models [212].

## Figures and Tables

**Figure 1 children-09-01834-f001:**
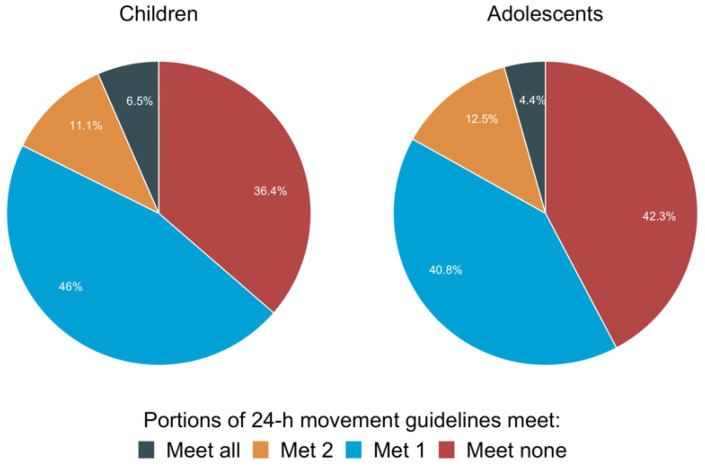
US National Children’s Health Survey is summarized regarding the distribution of 24 h movement guidelines (physical activity, screen time, and sleep duration recommendations) among young adults with ASD.

**Table 1 children-09-01834-t001:** Clinical effects of physical activity in children and adolescents with ASD.

Types of Physical Activity	Clinical Effect	Reference
Aquatic exercises	Improvements in sleep latency and duration	[79,80,81]
Walking/Running	Improvements in weight management and academic engagement	[82,83]
Cycling	Improvements in self-regulation and physical activity	[84]
Yoga and dance	Improvements in classroom behaviors, behavioral symptoms, and depression	[85,86]
Martial Arts	Improvements in stereotypy and communication deficits	[87,88,89]
Horseback riding	Improvements in executive function, irritability, and stereotypic behaviour	[90,91]
Fundamental movement skills	Improvements in anxiety symptoms	[92]
Basketball	Improvements in sleep quality	[93]
Football-based training	Improvements in executive function	[94]

**Table 2 children-09-01834-t002:** Gut microbiota in children and adolescents with ASD.

Biological Material	Microbes	Clinical Impact	References
Stool samples	ASD group had greater levels of *Bacteroidetes*, while the control group had greater levels of *Firmicutes.* Both *Actinobacterium* and *Proteobacterium* displayed smaller but significant differences. The prevalence of *Desulfovibrio* species and *Bacteroides vulgatus* in the stools of children with ASD is significantly higher than that in children without ASD.	Describing microbial signatures related with health	[114]
Ileal and cecal samples	Higher prevalence of *Sutterella* spp.	Children with ASD and gastrointestinal dysfunction have a significantly higher level of *Sutterella* present in their microbiota than children with only gastrointestinal dysfunction.	[128]
Stool samples	The number of *Sutterella* spp. in the feces of children with ASD is higher than that of controls.	The number of *Ruminococcus torques* is also higher in those children with ASD who have been diagnosed with a functional gastrointestinal disorder.	[130]
Stool samples	The *Firmicutes*/*Bacteroidetes* ratio significantly increased in patients with ASD due to a reduction in *Bacteroidetes*. Among the cohort with ASD, *Alistipes*, *Bilophila*, *Dialister*, *Parabacteroides*, and *Veillonella* decreased in relative abundance, while *Collinsella*, *Corynebacterium*, *Dorea*, and *Lactobacillus* increased. Children with ASD have a higher level of *Clostridium* cluster XVIII and *Escherichia*/*Shigella*. *Candida* abundances was also significantly different in ASD subjects compared to healthy subjects.	ASDs are associated with altered intestinal microbial communities at both the bacterial and fungal levels. This is not determined by the constipation status of ASD individuals, but rather by the ASD itself.	[127]
Stool samples	Children with ASD also had lower abundances of *Haemophilus parainfluenzae* and *Faecalibacterium prausnitzii* in their feces after multiple testing corrections.	The results obtained in this study suggest that children with ASD have different metabolite profiles in their feces.	[134]
Dental and salivary samples	ASD patients’ saliva and plaques contained increased levels of *Haemophilus* and *Streptococcus*. As a result, commensal bacteria *Selenomonas*, *Prevotella*, *Actinomyces*, *Fusobacterium*, and *Porphyromonas* were reduced in number.	Clinical indices, such as severity of disease and oral health, were also correlated with distinguishable bacteria.	[136]
Stool samples	Children with ASD have a gut microbiota that consists predominantly of *Bifidobacteraceae*, *Lactobacillaceae*, and *Veillonellaceae*; healthy children have a gut microbiota that consists primarily of *Prevotellaceae.*	In this study, differences in microbial community structure were identified between children with ASD and healthy children.	[131]
Stool samples	In the fecal microbiota of the ASD group, the *Bacteroidetes*/*Firmicutes* ratio increased significantly. The relative abundance of *Sutterella*, *Odoribacter*, and *Butyricimonas* was significantly higher in the ASD group as compared to the control group, whereas *Veillonella* and *Streptococcus* were significantly decreased.	ASD showed a positive correlation with periodontal disease and a negative correlation with type 1 diabetes in this microbe-disease network based on microbe similarity of diseases.	[139]
Stool samples	There was an increase in *Acidobacteria* but a decrease in *Ruminococcaceae*, *Eubacterium*, *Lachnospiraceae*, and *Erysipelotrichaceae* among ASD individuals.	ASD may be treatable by modulating the gut microbiota.	[140]
Stool samples	There was a much higher proportion of *Proteobacteria* and *Actinobacteria* in children with ASD, as well as *Bacilli*, *Actinobacteria*, *Erysipelotrichi*, and *Gammaproteobacteria* in the class of children with ASD.	In children with ASD, mixed bacterial and nutritional variables showed differential patterns.	[141]
Stool samples	In children with ASD, *Bacteroides*, *Coprococcus*, *Akkermansia*, and *Ruminococcus* spp. were more abundant and diverse.	Children with ASD have abnormal eating habits, as well as common gastrointestinal symptoms that are associated with nutritional differences.	[132]
Saliva and stool samples	*Bacilli* was significantly higher in the gut of ASD individuals compared to controls.	ASD and comorbid conditions can be diagnosed and treated using microbial markers.	[135]

Abbreviations. ASD, autism spectrum disorder.

## Data Availability

Not applicable.

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
