# Peer review of "Physical Activity, Gut Microbiota, and Genetic Background for Children and Adolescents with Autism Spectrum Disorder"

_children, 2022, doi:10.3390/children9121834_

Round 1

Reviewer 1 Report

In the study titled Physical Activity, Gut Microbiota, and Genetic Background for 2 Children and Adolescents with Autism Spectrum Disorder, carried out Julio Plaza-Diaz et al., a review of the literature on the problem of the physiopathology of autism in children with a search depth of 1985-2022 is presented. The influence of physical activity on gastrointestinal problems and improvement of the microbiota is considered, which is of practical importance and can be recommended as a non-drug method for improving symptoms associated with autism spectrum disorders (ASD) in children. The issues of the gut-microbiota-brain connection are widely covered, as well as data on genome-wide screenings for ASD. Cause-and-effect relationships of single nucleotide variations and the gut microbiome in people with ASD have been determined.

The work is well presented and may be of interest to fundamentalists and clinical specialists in the field of ASD.

However, there are a number of significant comments on the work for its improvement, related to the clarity of the presentation of the material:

1.       In Section 2. Physical activity with a special emphasis on ASD, data from the U.S. National Children's Health Survey on the distribution of physical activity among young adults with ASD should be supplemented with a figure (pie chart) for better clarity and reproducibility of the material (lines 114-122). 

2.       In Section 2.1. Physical exercise interventions for children and adolescents with ASD, some of the material on physical activity among children with ASD can be presented as a table with the following columns: types of physical activity, clinical effect, references to the literature source, which will also improve the article.

3.       In section 3: Gut microbiota in children and adolescents with ASD, the descriptive material can also be supplemented with a table containing the following columns: type of biological material, strains of microorganisms, clinical manifestations, references to the literature source.

4.       Section 5. Conclusions further reflections on genetics in children with ASD and their relationship to the gut microbiota are required.

Author Response

Dear Ms. Catherine Nie

Section Managing Editor,

Thank you for providing us the opportunity to submit a revised version of our editorial entitled Physical Activity, Gut Microbiota, and Genetic Background for Children and Adolescents with Autism Spectrum Disorderto Children.

We thank the reviewers for their thoughtful comments and suggestions regarding our manuscript. We have taken into account all of the comments from reviewers and also the repetition rate report and incorporated them into the revised manuscript. An itemized point-by-point response to the comments from the reviewers is provided below in response to the changes made to the original document. Line numbers are taken from the track-changed manuscript, however they may change during the conversion to PDF.

COMMENTS FROM REVIEWER 1

Comment #1

In the study titled Physical Activity, Gut Microbiota, and Genetic Background for Children and Adolescents with Autism Spectrum Disorder, carried out Julio Plaza-Diaz et al., a review of the literature on the problem of the physiopathology of autism in children with a search depth of 1985-2022 is presented. The influence of physical activity on gastrointestinal problems and improvement of the microbiota is considered, which is of practical importance and can be recommended as a non-drug method for improving symptoms associated with autism spectrum disorders (ASD) in children. The issues of the gut-microbiota-brain connection are widely covered, as well as data on genome-wide screenings for ASD. Cause-and-effect relationships of single nucleotide variations and the gut microbiome in people with ASD have been determined. The work is well presented and may be of interest to fundamentalists and clinical specialists in the field of ASD.

Response: Thanks to the reviewer for his/her kind comment about our manuscript.

Comment #2

However, there are a number of significant comments on the work for its improvement, related to the clarity of the presentation of the material: In Section 2. Physical activity with a special emphasis on ASD, data from the U.S. National Children's Health Survey on the distribution of physical activity among young adults with ASD should be supplemented with a figure (pie chart) for better clarity and reproducibility of the material (lines 114-122).

Response: Thanks to the reviewer for his/her kind comment about our manuscript. Using the reviewer’s comment, the pie chart was added as Figure 1.

Comment #3

In Section 2.1. Physical exercise interventions for children and adolescents with ASD, some of the material on physical activity among children with ASD can be presented as a table with the following columns: types of physical activity, clinical effect, references to the literature source, which will also improve the article.

Response: Thanks to the reviewer for his/her kind comment about our manuscript. Using the reviewer’s comment, the Table 1 was added.

Comment #4

In section 3: Gut microbiota in children and adolescents with ASD, the descriptive material can also be supplemented with a table containing the following columns: type of biological material, strains of microorganisms, clinical manifestations, references to the literature source.

Response: Thanks to the reviewer for his/her kind comment about our manuscript. Using the reviewer’s comment, the Table 2 was added.

Comment #5

Section 5. Conclusions further reflections on genetics in children with ASD and their relationship to the gut microbiota are required.

Response: Thanks to the reviewer for his/her kind comment about our manuscript. Using the reviewer’s comment, the conclusion was modified, and now states (page 14, lines 1807-1809), “An increasing amount of research is being conducted on how host genetics interact with the gut microbiome [212]. This research is in the context of complex human diseases, with previous evidence coming primarily from animal models [212].”

Reviewer 2 Report

The manuscript addressed very important topic, which is gut microbiome in ASD subjects, however, the title did not reflect well the content of the paper. I could not find the link between physical activity and  gut microbiota in ASD. The focus was more on gut microbiome in general, and also genetics of ASD.

I suggest adding a section on the relation between microbiome and physical activity, and if any studies were done in relation to ASD. Also, please try to connect the sections in a better way, for example, can the genetics of ASD be better connected to microbiome , I could see a short note on this in the last paragraph of section 4, but to the best of my knowledge, there are studies connecting the genetics of ASD to gut microbiome

Some minor edits:

Line 210: Change antibiotic to antibiotics

Line 211: why ADS children are susceptible to ear infections particularly? I suggest making the sentence specific to gut, remove ear infections

Line 228 : change More to more

Line 229: not clear : “than is typical these may be correlated” , use the term typically developed children , and please revise the sentence

Line 232-233 : “had a decrease” must be changed to decreased , “had an increase” must be changed to increased

 Line 249: Prevotellaceae , P in the name must be italicized

Author Response

Dear Ms. Catherine Nie

Section Managing Editor,

Thank you for providing us the opportunity to submit a revised version of our editorial entitled Physical Activity, Gut Microbiota, and Genetic Background for Children and Adolescents with Autism Spectrum Disorderto Children.

We thank the reviewers for their thoughtful comments and suggestions regarding our manuscript. We have taken into account all of the comments from reviewers and also the repetition rate report and incorporated them into the revised manuscript. An itemized point-by-point response to the comments from the reviewers is provided below in response to the changes made to the original document. Line numbers are taken from the track-changed manuscript, however they may change during the conversion to PDF.

COMMENTS FROM REVIEWER 2

Comment #1

The manuscript addressed very important topic, which is gut microbiome in ASD subjects, however, the title did not reflect well the content of the paper. I could not find the link between physical activity and  gut microbiota in ASD. The focus was more on gut microbiome in general, and also genetics of ASD. I suggest adding a section on the relation between microbiome and physical activity, and if any studies were done in relation to ASD. Also, please try to connect the sections in a better way, for example, can the genetics of ASD be better connected to microbiome , I could see a short note on this in the last paragraph of section 4, but to the best of my knowledge, there are studies connecting the genetics of ASD to gut microbiome

Response: Thanks to the reviewer for his/her kind comment about our manuscript. New information was added about physical activity and gut microbiota and the manuscript now states (page 6, lines 1138-1140), “Exercise and physical activity have also been shown to change the composition of the gut microbiota, something which is associated with improved energy homeostasis and regulation (for a comprehensive revision, please refer to [42,104]) [42,104-106.”, about the connections, the manuscript was modified and the last request about ASD genetics and microbiome, the manuscript now states (page 10, lines 1407-1413), “Recent studies have demonstrated that host genetics influence both the composition of the overall microbiome as well as the composition of individual bacterial species [152]. In Knight et al., changes in Enterobacteriaceae abundance were associated with Crohn's risk variants within the NOD2 gene [153]. In a genome-wide association study of host genetics and microbiome in healthy individuals, it has been found that a loss-of-function variant in the fucosyl transferase 2 gene and a variant conferring hypolactasia near the lactase gene are associated with stool abundance of B. longum [154].”

Comment #2

Some minor edits, Line 210: Change antibiotic to antibiotics

Response: Using the reviewer’s comment. The sentence was modified.

Comment #3

Line 211: why ADS children are susceptible to ear infections particularly? I suggest making the sentence specific to gut, remove ear infections

Response: Using the reviewer’s comment. The sentence was modified and now states (page 6, lines 1161-1164), “Given that antibiotic usage affects the health and composition of the gut microbiome [113] and the fact that children with ASD are frequently prescribed antibiotics due to their susceptibility to gastrointestinal infections [114], the study of the gut microbiota in ASD represents a compelling research topic.”.

Comment #4

Line 228 : change More to more

Response: Using the reviewer’s comment. The sentence was modified.

Comment #5

Line 229: not clear : “than is typical these may be correlated” , use the term typically developed children , and please revise the sentence

Response: Using the reviewer’s comment. The sentence was modified and now states (page 6, lines 1187-1189), “Previous studies have shown Sutterella wadsworthensis to be associated with gastro-intestinal infections; interestingly, ASD children with and without digestive disturbances, have a significantly higher prevalence of Sutterella species [128-130]”

Comment #6

Line 232-233 : “had a decrease” must be changed to decreased , “had an increase” must be changed to increased

Response: Using the reviewer’s comment. The sentence was modified.

Comment #7

Line 249: Prevotellaceae , P in the name must be italicized

Response: Using the reviewer’s comment. The sentence was modified.

Reviewer 3 Report

This review articles is very informative, addressing the the role of physical activity, microbiota in ASD. It is imperative to study role of these two parameters considering the recent evidences that showed autistic subjects harbor an altered gut microbiota which can influence the host immune system and prone to various infections (GI and skin).  Authors did a very good job by shading light on these important aspects with ASD. However, there are some minor limitations with this review. Here are my overall comments.

Major comments

1. Authors used other systematic review as references on multiple instances, please use original studies where ever possible. 

2. This review lacks the role of maternal microbiota and mode of delivery Normal delivery vs C section in ASD.

3. Fatal neurodevelopment and early diagnosis of ASD

4. In therapeutic section, please include FMT (fecal microbiota transplantation) could be useful in ASD as it showed promising results in C. difficile infections. 

Minor comments

1. Please define alpha and beta diversity once in MS.

2. Line 48: ASD is more prevalent in males compared to females in the introduction part but nothing is explain further in the MS in this regard.

3. Line 108-109: Authors wants reviewer to add this information? Please delete this.

Author Response

Dear Ms. Catherine Nie

Section Managing Editor,

Thank you for providing us the opportunity to submit a revised version of our editorial entitled Physical Activity, Gut Microbiota, and Genetic Background for Children and Adolescents with Autism Spectrum Disorderto Children.

We thank the reviewers for their thoughtful comments and suggestions regarding our manuscript. We have taken into account all of the comments from reviewers and also the repetition rate report and incorporated them into the revised manuscript. An itemized point-by-point response to the comments from the reviewers is provided below in response to the changes made to the original document. Line numbers are taken from the track-changed manuscript, however they may change during the conversion to PDF.

COMMENTS FROM REVIEWER 3

Comment #1

This review articles is very informative, addressing the the role of physical activity, microbiota in ASD. It is imperative to study role of these two parameters considering the recent evidences that showed autistic subjects harbor an altered gut microbiota which can influence the host immune system and prone to various infections (GI and skin).  Authors did a very good job by shading light on these important aspects with ASD.

Response: Thanks to the reviewer for his/her kind comment about our manuscript.

Comment #2

However, there are some minor limitations with this review. Here are my overall comments.

Major comments, Authors used other systematic review as references on multiple instances, please use original studies where ever possible.

Response: Using the reviewer’s comment, the systematic review was changed for original where ever possible.

Comment #3

This review lacks the role of maternal microbiota and mode of delivery Normal delivery vs C section in ASD.

Response: Using the reviewer’s comment, the information was added in the manuscript, and now states (page 2, lines 235-247) “It is becoming increasing recognized that the mode of newborn delivery, vaginal or Cesarean section, can have a profound impact on both early and late outcomes in the life of a child, chiefly because of the significant differences observed in the newborn microbiome [23]. Infections, allergies, and inflammatory disorders are more likely to occur following Cesarean delivery [24-27]. ASD has been extensively studied in this regard, with many studies showing Cesarean section, when compared with natural birth, confers an increased risk of developing ASD [28-31] including a Canadian study showing a 1.23 times greater risk [32]. A 2016 cohort study of 5 million children in Norway, Sweden, Denmark, Finland, and Australia showed this number to be 1.26 for emergency or planned Cesarean sections [33]. However, such an association was not detected when sibling controls were used in another study suggesting the existence of confounding familial genetic and/or environmental factors [34]. There is clearly a need for further investigation of the relationship between Caesarean delivery and ASD.”

Comment #4

Fetal neurodevelopment and early diagnosis of ASD

Response: Using the reviewer’s comment, the information was added in the manuscript, and now states (page 2, lines 229-234) “In addition to DNA variants, there are several heritable and nonheritable risk fac-tors associated with ASD included increased postnatal head growth, brain volume anomalies, neuroanatomical defects in the amygdala, immune system dysregulation, serotonergic systems anomalies, and distinct neuropeptides or neurotrophins [20]. Putative neuropathological processes are proposed to begin in utero [20] including antenatal dysregulation of cortical layer formation and neuronal differentiation [21,22].

Comment #5

In therapeutic section, please include FMT (fecal microbiota transplantation) could be useful in ASD as it showed promising results in C. difficile infections.

Response: Thanks to the reviewer for his/her, we stated that information in the manuscript with another description, as “microbiota transplant therapy”, at the end of the paragraph we have summarizes the promising results. Now the manuscript states (pages 11-12, lines 1681-1698) “In contrast, an open-label study with two-year follow-up, treating patients with fecal microbiota transfer therapy (microbiota transplantation) proved relatively effective and safe, with significant reductions in gastrointestinal disorders and ASD symptoms as well as  changes in the gut microbiota, and an increase in microbial diversity all observed [171]. The documented significant differences in plasma metabolites were to some extent corrected, approximating those found in healthy children [172]. As a result of microbiota transfer therapy, overall bacterial diversity also increased, with greater abundance  of Bifidobacterium, Prevotella, and Desulfovibrio. The changes persisted for at least 8 weeks after treatment cessation [173].

Another report suggests the impact of microbiota transplant therapy on gut micro-biota may be related, at least in part, to the presence of Eubacterium coprostanoligene [174]. In another study, a significant reduction in both constipation and abnormal feces, as well as white blood cells, and globulin levels, was observed after microbiota trans-plant therapy with a positive correlation between size effect and number of treatments [175].

The use of microbiota transplant therapy may thus be an effective therapeutic strategy in the treatment of both digestive and behavioral symptoms in patients with ASD, and also has shown favorable results in the treatment of certain infections, such as C. difficile [176].”

Comment #6

Minor comments, Please define alpha and beta diversity once in MS.

Response: Using the reviewer’s comment, the information was added in the manuscript, and now states (page 6, lines 1167-1172) “ASD intestinal dysbiosis is characterized by persistently reduced alpha diversity (de-fined as mean diversity of species at different locations within a local scale [117] while a beta diversity can be defined as the ratio between the mean diversity of species at regional and local levels), the presence of immature microbes, an altered composition of 20 operational taxonomic units, reduced detection rates of taxon, and 325 metabolic functions that are deregulated [118].”

Comment #7

Line 48: ASD is more prevalent in males compared to females in the introduction part but nothing is explain further in the MS in this regard.

Response: Using the reviewer’s comment, the information was added in the manuscript, and now states (pages 1-2, lines 45-218) “The male-to-female ratio of ASD has long been thought to be 4:1 [7] although a recent systematic review has found it may be closer to 3:1 [8]. One reason for this ostensible male predominance is the existence of an apparent diagnostic gender bias, girls who meet ASD criteria are disproportionately less likely to be diagnosed with the condition [8]. Environmental and genetic factors also contribute to a child's likelihood of developing ASD [6], including sex-linked genetic factors (e.g., the X chromosome gene protective effect) and hormonal factors (e.g., prenatal hormones), which are proposed to attenuate the risk in females and increase it in males [9-17].“

Comment #8

Line 108-109: Authors wants reviewer to add this information? Please delete this.

Response: Thanks to the reviewer for his/her comment, the sentence was deleted.

Round 2

Reviewer 2 Report

All the comments were addressed and the revised version is very clear and better with the added information